# Nonconvex Theory of $M$-estimators with Decomposable Regularizers

**Weiwei Liu** [1]

## Abstract

High-dimensional inference addresses scenarios where the dimension of the data approaches, or even surpasses, the sample size. In these settings, the regularized $M$-estimator is a common technique for inferring parameters. (Negahban et al., 2009) establish a unified framework for establishing convergence rates in the context of high-dimensional scaling, demonstrating that estimation errors are confined within a restricted set, and revealing fast convergence rates. The key assumption underlying their work is the convexity of the loss function. However, many loss functions in high-dimensional contexts are nonconvex. This leads to the question: if the loss function is nonconvex, do estimation errors still fall within a restricted set? If yes, can we recover convergence rates of the estimation error under nonconvex situations? This paper provides affirmative answers to these critical questions.

## 1. Introduction

Recently, the emerging trends of high feature dimensionality have been studied in (Mitchell et al., 2004; Fan et al., 2009; Zhai et al., 2014; Liu & Tsang, 2015; 2017; Liu et al., 2017b; 2019b;a; Liu & Tsang, 2016). For example, quintillion bytes of data are generated by the Web on a daily basis (Zikopoulos et al., 2012; Zhai et al., 2014; Liu et al., 2017a; 2018; Liu & Shen, 2019; Gong et al., 2023b;a; 2021). This surge in data has led to situations where the dimensionality of features ($d$) approaches or even surpasses the size of the sample size ($n$). In such regimes, classical "large $n$, fixed $d$" theory often fails to provide useful predictions, and the performance of standard methods will also be significantly degraded. Thus, it is imperative to develop new theory as

[1]School of Computer Science, National Engineering Research Center for Multimedia Software, Institute of Artificial Intelligence and Hubei Key Laboratory of Multimedia and Network Communication Engineering, Wuhan University, Wuhan, China. Correspondence to: Weiwei Liu <liuweiwei863@gmail.com>.

*Proceedings of the 42$^{nd}$ International Conference on Machine Learning*, Vancouver, Canada. PMLR 267, 2025. Copyright 2025 by the author(s).

well as new methods in high-dimensional regimes.

The regularization $M$-estimator technique has gained significant attention as a means of estimating model parameters within high-dimensional contexts (Liu, 2023). (Negahban et al., 2009) introduce a comprehensive framework for the regularized $M$-estimator. This framework not only consolidates and extends prior findings but also introduces fresh insights into consistency and convergence rates within the high-dimensional situation. Their work is further documented in the seminal textbook on high-dimensional statistics authored by (Wainwright, 2019). Specifically, (Negahban et al., 2009) develop two pivotal properties of the regularized $M$-estimator: Firstly, they establish the decomposability property of the regularizer; Secondly, they introduce the concept of restricted strong convexity for the loss function. When the loss function is convex and the regularizer exhibits decomposability, their analysis reveals that the error vector between any global optimum of the regularized $M$-estimator and the unknown ground truth parameter falls within a tightly constrained set. Additionally, in cases where the loss function demonstrates convexity and complies with the conditions of restricted strong convexity, and the regularizer is decomposable, one of their principal theorems establishes the convergence rates for the estimation error under high-dimensional scaling.

The principal premise underpinning their primary findings relies on the convexity of the loss function. However, in high-dimensional settings, many loss functions exhibit nonconvex characteristics. For example, mixture models and high-dimensional sparse linear regression models with noisy and/or missing data are nonconvex, which will be described in details in subsequent sections. This motivates us to ask the following questions:

1. When the loss function is nonconvex, do estimation errors still fall within a constrained set?

2. If yes, can we recover the convergence rates of the estimation error under nonconvex regimes?

This paper provides affirmative answers to these questions.

**Contributions**. This paper advances the theory of regularized $M$-estimator with decomposable regularizers from convex to nonconvex. Our main results show that the esti-

mation errors still lie in a restricted set and we can recover the convergence rates of the estimation error when the loss function is nonconvex. Moreover, we apply our main results to two nonconvex models: corrected linear regression and $\ell_1$-penalized Lasso estimator. Our key technical analysis of two examples is to prove that with high probability, a form of the restricted strong convexity (RSC) condition and dual norm bound hold.

**Related Work.** (Loh & Wainwright, 2015) is a seminal work, which presents the nonconvex results of regularized $M$-estimator. (Loh & Wainwright, 2015) focus on the non-convex regularizers, which are defined in Assumption 1 of (Loh & Wainwright, 2015). Specially, nonconvex SCAD and MCP regularizers are studied by (Loh & Wainwright, 2015). This work focuses on the decomposable regularizers, which are defined in Definition 2.1 of our paper. Becasue the definitions of nonconvex regularizers and decomposable regularizers are totally different, the main results of (Loh & Wainwright, 2015) can not be extended to decomposable regularizers.

## 2. Problem Formulation

Let $Z$ be a random variable with distribution $\mathbb{P}$ taking values in a set $\mathcal{Z}$. Suppose that we observe a collection of $n$ samples $Z_1^n = (Z_1, \ldots, Z_n)$, where each sample $Z_i$ is drawn in an independent and identically distributed (i.i.d.) manner from $\mathbb{P}$. Consider a loss function $\mathcal{L}_n : \mathbb{R}^d \times \mathcal{Z}^n \to \mathbb{R}$. The value $\mathcal{L}_n(\theta; Z_1^n)$ measures the fit between a parameter vector $\theta \in \mathbb{R}^d$ and the observed data $Z_1^n$. Its expectation defines the population cost function:

$$\bar{\mathcal{L}}(\theta) := E[\mathcal{L}_n(\theta; Z_1^n)]$$

Suppose that the cost function has an additive decomposition of the form $\mathcal{L}_n(\theta; Z_1^n) = 1/n \sum_{i=1}^n \mathcal{L}(\theta; Z_i)$, where $\mathcal{L} : \mathbb{R}^d \times \mathcal{Z} \to \mathbb{R}$ is the cost defined on a single sample. Then, the population cost function does not depend on the sample size $n$. The target parameter vector $\theta^* \in \mathbb{R}^d$ is defined as the minimum of the population cost function:

$$\theta^* = \arg \min_{\theta \in \mathbb{R}^d} \bar{\mathcal{L}}(\theta)$$

Our goal is to estimate the unknown parameter $\theta^*$ from the observed samples $Z_1^n$. In order to do so, we combine the empirical cost function with a regularizer or penalty function $\Phi : \mathbb{R}^d \to \mathbb{R}$. The regularized $M$-estimator is based on solving the following optimization problem:

$$\hat{\theta} \in \arg \min_{\theta \in \mathbb{R}^d} \{ \mathcal{L}_n(\theta; Z_1^n) + \lambda_n \Phi(\theta) \} \quad (1)$$

where $\lambda_n > 0$ is a user-defined regularization parameter. For ease of notation, we adopt $\mathcal{L}_n(\theta)$ as a shorthand for $\mathcal{L}_n(\theta; Z_1^n)$ in the sequel. We also adopt the same notation for the derivatives of the empirical cost function. Throughout the paper, we assume that the loss function is differentiable, and that the regularizer $\Phi$ is a norm.

### 2.1. Decomposable Regularizers

The decomposability of the regularizer is introduced by (Negahban et al., 2009; Wainwright, 2019) to analyze the estimation error bound for convex loss functions. We assume that the space $\mathbb{R}^d$ is endowed with an inner product $\langle \cdot, \cdot \rangle$, and we use $|| \cdot ||$ to denote the norm induced by this inner product.

The notion of a decomposable regularizer is defined in terms of a pair of subspaces $\mathbb{M} \subseteq \bar{\mathbb{M}}$ of $\mathbb{R}^d$. We define the orthogonal complement of the space $\bar{\mathbb{M}}$ as $\bar{\mathbb{M}}^\perp := \{ v \in \mathbb{R}^d | \langle u, v \rangle = 0, \forall u \in \bar{\mathbb{M}} \}$. The definition of the decomposability is provided as below:

**Definition 2.1** (Decomposability). Given a pair of subspaces $\mathbb{M} \subseteq \bar{\mathbb{M}}$, a norm-based regularizer $\Phi$ is decomposable with respect to $(\mathbb{M}, \bar{\mathbb{M}}^\perp)$ if

$$\Phi(\alpha + \beta) = \Phi(\alpha) + \Phi(\beta), \forall \alpha \in \mathbb{M}, \forall \beta \in \bar{\mathbb{M}}^\perp \quad (2)$$

The $\ell_1$ norm is the canonical example of a decomposable norm. There are some other norms that also share this property, such as the group Lasso norm and the nuclear norm.

Given a vector $\theta \in \mathbb{R}^d$ and a subspace $\mathbb{S}$ of $\mathbb{R}^d$, we use $\theta_{\mathbb{S}}$ to denote the projection of $\theta$ onto $\mathbb{S}$: $\theta_{\mathbb{S}} := \arg \min_{\tilde{\theta} \in \mathbb{S}} ||\tilde{\theta} - \theta||^2$. Consider any norm $\Phi : \mathbb{R}^d \to \mathbb{R}$, its dual norm is defined as $\Phi^*(v) := \sup_{\Phi(u) \leq 1} \langle u, v \rangle$. We define $\mathbb{G}(\lambda_n) := \{ \Phi^*(\nabla \mathcal{L}_n(\theta^*)) \leq \lambda_n/2 \}$.

The decomposability plays an important role in the $M$-estimation. The following Proposition introduced by (Wainwright, 2019) shows that the error vector $\hat{\theta} - \theta^*$ between any global optimum of the optimization problem (1) and the unknown parameter $\theta^*$ lie in a very restricted set.

**Proposition 2.2.** *Let $\mathcal{L}_n : \mathbb{R}^d \times \mathcal{Z}^n \to \mathbb{R}$ be a convex function, let the regularizer $\Phi : \mathbb{R}^d \to [0, \infty)$ be a norm, and consider a subspace pair $(\mathbb{M}, \bar{\mathbb{M}}^\perp)$ over which $\Phi$ is decomposable. Then conditioned on the event $\mathbb{G}(\lambda_n)$, the error $\hat{\Delta} = \hat{\theta} - \theta^*$ belongs to the set*

$$\mathbb{C}_{\theta^*}(\mathbb{M}, \bar{\mathbb{M}}^\perp) := \{ \Delta \in \mathbb{R}^d | \Phi(\Delta_{\bar{\mathbb{M}}^\perp}) \leq 3\Phi(\Delta_{\bar{\mathbb{M}}}) \\ + 4\Phi(\theta^*_{\mathbb{M}^\perp}) \} \quad (3)$$

This key property of decomposability is based on the convexity of the loss function, which prompts us to ask the first question:

*Whether the results of Proposition 2.2 still hold if the loss function is nonconvex?*

This paper provides an affirmative answer to this question.

## 2.2. Restricted Strong Convexity

Given any differentiable loss function, the first-order Taylor-series error is defined as:

$$\mathcal{E}_n(\Delta) := \mathcal{L}_n(\theta^* + \Delta) - \mathcal{L}_n(\theta^*) - \langle \nabla \mathcal{L}_n(\theta^*), \Delta \rangle$$

This error term is always guaranteed to be nonnegative if the loss function is convex. The strong convexity is used to ensure that a function is not too flat and is defined as $\mathcal{E}_n(\Delta) \geq \kappa/2||\Delta||^2$ for all $\Delta \in \mathbb{R}^d$. (Wainwright, 2019) show that the strong convexity cannot hold for a generic high-dimensional problem. However, for decomposable regularizers, Proposition 2.2 shows that the error vector must lie within a restricted set, and (Wainwright, 2019) use this fact to define the notion of restricted strong convexity as follows.

**Definition 2.3** (Restricted Strong Convexity (Wainwright, 2019)). For a given norm $|| \cdot ||$ and regularizer $\Phi(\cdot)$, the loss function satisfies a restricted strong convexity (RSC) condition with radius $R > 0$, curvature $\kappa > 0$ and tolerance $\tau_n^2$ if

$$\mathcal{E}_n(\Delta) \geq \kappa/2||\Delta||^2 - \tau_n^2 \Phi^2(\Delta), \forall \Delta \in \mathbb{B}(R) \quad (4)$$

where $\mathbb{B}(R)$ is the unit ball defined by the given norm $|| \cdot ||$.

The following results involve the subspace Lipschitz constant.

**Definition 2.4** (Subspace Lipschitz Constant). For any subspace $\mathbb{S}$ of $\mathbb{R}^d$, the subspace Lipschitz constant with respect to the pair $(\Phi, || \cdot ||)$ is given by

$$\Psi(\mathbb{S}) := \sup_{u \in \mathbb{S} \setminus \{0\}} \frac{\Phi(u)}{||u||} \quad (5)$$

Consider the following assumptions:

**Assumption 2.5.** The loss function is convex, and satisfies the local RSC condition (4) with curvature $\kappa$, radius $R$ and tolerance $\tau_n^2$ with respect to an inner-product induced norm $|| \cdot ||$.

**Assumption 2.6.** There is a pair of subspaces $\mathbb{M} \subseteq \bar{\mathbb{M}}$ such that the regularizer decomposes over $(\mathbb{M}, \bar{\mathbb{M}}^{\perp})$.

Based on these assumptions, (Wainwright, 2019) present the key bounds for the general models. We first define the quantity involved in the bounds:

$$\varepsilon_n^2(\bar{\mathbb{M}}, \mathbb{M}^{\perp}) := 9 \frac{\lambda_n^2}{\kappa^2} \Psi^2(\bar{\mathbb{M}}) + \frac{8}{\kappa} \{ \lambda_n \Phi(\theta_{\mathbb{M}^{\perp}}^*) + 16 \tau_n^2 \Phi^2(\theta_{\mathbb{M}^{\perp}}^*) \} \quad (6)$$

**Theorem 2.7** (Bounds for general models). *Under Assumptions 2.5 and 2.6, consider the regularized $M$-estimator (1) conditioned on the event $\mathbb{G}(\lambda_n)$,*

*(i) Any optimal solution satisfies the bound*

$$\Phi(\hat{\theta} - \theta^*) \leq 4 \{ \Psi(\bar{\mathbb{M}}) ||\hat{\theta} - \theta^*|| + \Phi(\theta_{\mathbb{M}^{\perp}}^*) \} \quad (7)$$

*(ii) For any subspace pair $(\bar{\mathbb{M}}, \mathbb{M}^{\perp})$ such that $\tau_n^2 \Psi^2(\bar{\mathbb{M}}) \leq \kappa/64$ and $\varepsilon_n(\bar{\mathbb{M}}, \mathbb{M}^{\perp}) \leq R$, we have*

$$||\hat{\theta} - \theta^*||^2 \leq \varepsilon_n^2(\bar{\mathbb{M}}, \mathbb{M}^{\perp}) \quad (8)$$

The proof of Theorem 2.7 in (Wainwright, 2019) shows that (7) does not depend on the convexity of the loss function, while (8) depends on the convex assumption of the loss function, which prompts us to ask the second question:

> *Can we recover the convergence rates of the error $||\hat{\theta} - \theta^*||^2$ (8) if the loss function is nonconvex?*

This paper provides an affirmative answer to this question.

## 3. Main Results

This section turns to the statements of our main statistical guarantees, which applies to any vector $\tilde{\theta} \in \mathbb{R}^d$ that satisfies the first-order necessary conditions to be a local minimum of (1):

$$\langle \nabla \mathcal{L}_n(\tilde{\theta}) + \lambda_n \nabla \Phi(\tilde{\theta}), \theta - \tilde{\theta} \rangle \geq 0, \theta \in \mathbb{R}^d \quad (9)$$

Such vectors $\tilde{\theta}$ satisfying the condition (9) are also known as stationary points.

Our main statistical results are based on the weaker RSC condition (Loh & Wainwright, 2015) than (4), which shows below:

$$\langle \nabla \mathcal{L}_n(\theta^* + \Delta) - \nabla \mathcal{L}_n(\theta^*), \Delta \rangle \geq \kappa/2||\Delta||^2 - \tau_n^2 \Phi^2(\Delta), \forall \Delta \in \mathbb{B}(R) \quad (10)$$

For any vector $\hat{\theta} \in \mathbb{R}^d$ satisfies the first-order necessary conditions (9), we define $\tilde{\mathbb{G}}(\lambda_n) := \{ \Phi^*(\nabla \mathcal{L}_n(\hat{\theta})) \leq \lambda_n/2 \}$.

**Theorem 3.1.** *Let the regularizer $\Phi : \mathbb{R}^d \to [0, \infty)$ be a norm, and consider a subspace pair $(\mathbb{M}, \bar{\mathbb{M}}^{\perp})$ over which $\Phi$ is decomposable. Consider any vector $\hat{\theta} \in \mathbb{R}^d$ satisfies the first-order necessary conditions (9), conditioned on the event $\tilde{\mathbb{G}}(\lambda_n)$, the error $\hat{\Delta} = \hat{\theta} - \theta^*$ belongs to the set*

$$\mathbb{C}_{\theta^*}(\mathbb{M}, \bar{\mathbb{M}}^{\perp}) := \{ \Delta \in \mathbb{R}^d | \Phi(\Delta_{\bar{\mathbb{M}}^{\perp}}) \leq 3\Phi(\Delta_{\bar{\mathbb{M}}}) + 4\Phi(\theta_{\mathbb{M}^{\perp}}^*) \} \quad (11)$$

**Remark.** Theorem 3.1 shows that the results of the Proposition 2.2 in (Wainwright, 2019) still hold for any stationary points.

The proof of Theorem 3.1 uses the following Lemma.

**Lemma 3.2.** *For any decomposable regularizer and parameters $\theta^*$ and $\Delta$, we have*

$$\Phi(\Delta + \theta^*) - \Phi(\theta^*) \geq \Phi(\Delta_{\bar{\mathbb{M}}^\perp}) - \Phi(\Delta_{\bar{\mathbb{M}}}) - 2\Phi(\theta^*_{\mathbb{M}^\perp}) \tag{12}$$

The proof of Lemma 3.2 can be found in (Wainwright, 2019).

*Proof.* (of Theorem 3.1) Suppose vector $\hat{\theta} \in \mathbb{R}^d$ satisfies the first-order necessary conditions (9). Since $\theta^*$ is feasible, (9) implies that

$$\langle \nabla \mathcal{L}_n(\hat{\theta}) + \lambda_n \nabla \Phi(\hat{\theta}), \theta^* - \hat{\theta} \rangle \geq 0 \tag{13}$$

Rearranging (13) yields

$$\langle \nabla \mathcal{L}_n(\hat{\theta}), \hat{\Delta} \rangle + \langle \lambda_n \nabla \Phi(\hat{\theta}), \hat{\Delta} \rangle \leq 0 \tag{14}$$

Applying the Hölder's inequality with the regularizer and its dual, we have

$$\langle \nabla \mathcal{L}_n(\hat{\theta}), \hat{\Delta} \rangle \geq -\Phi^*(\nabla \mathcal{L}_n(\hat{\theta}))\Phi(\hat{\Delta})$$
$$\geq -\frac{\lambda_n}{2}(\Phi(\hat{\Delta}_{\bar{\mathbb{M}}}) + \Phi(\hat{\Delta}_{\bar{\mathbb{M}}^\perp})) \tag{15}$$

where the final step uses the assumed bound $\Phi^*(\nabla \mathcal{L}_n(\hat{\theta})) \leq \lambda_n/2$ and the triangle inequality. Using the convexity of the regularizer, we have

$$\Phi(\hat{\Delta} + \theta^*) - \Phi(\theta^*) \leq \langle \nabla \Phi(\hat{\Delta} + \theta^*), \hat{\Delta} \rangle \tag{16}$$

Combining (16) with Lemma 3.2, we obtain

$$\langle \lambda_n \nabla \Phi(\hat{\theta}), \hat{\Delta} \rangle = \langle \lambda_n \nabla \Phi(\hat{\Delta} + \theta^*), \hat{\Delta} \rangle$$
$$\geq \lambda_n(\Phi(\hat{\Delta} + \theta^*) - \Phi(\theta^*))$$
$$\geq \lambda_n(\Phi(\hat{\Delta}_{\bar{\mathbb{M}}^\perp}) - \Phi(\hat{\Delta}_{\bar{\mathbb{M}}}) - 2\Phi(\theta^*_{\mathbb{M}^\perp})) \tag{17}$$

Combining (14), (15) and (17) yields

$$0 \geq \lambda_n(\Phi(\hat{\Delta}_{\bar{\mathbb{M}}^\perp}) - \Phi(\hat{\Delta}_{\bar{\mathbb{M}}}) - 2\Phi(\theta^*_{\mathbb{M}^\perp}))$$
$$- \frac{\lambda_n}{2}(\Phi(\hat{\Delta}_{\bar{\mathbb{M}}}) + \Phi(\hat{\Delta}_{\bar{\mathbb{M}}^\perp}))$$
$$= \frac{\lambda_n}{2}(\Phi(\hat{\Delta}_{\bar{\mathbb{M}}^\perp}) - 3\Phi(\hat{\Delta}_{\bar{\mathbb{M}}}) - 4\Phi(\theta^*_{\mathbb{M}^\perp})) \tag{18}$$

This completes the proof of Theorem 3.1. □

**Theorem 3.3.** *Let the regularizer $\Phi : \mathbb{R}^d \to [0, \infty)$ be a norm, and consider a subspace pair $(\mathbb{M}, \bar{\mathbb{M}}^\perp)$ over which $\Phi$ is decomposable. Suppose the loss function satisfies the RSC condition (10). Consider any vector $\hat{\theta} \in \mathbb{R}^d$ satisfies the first-order necessary conditions (9), conditioned on the event $\tilde{\mathbb{G}}(\lambda_n)$, for any subspace pair $(\bar{\mathbb{M}}, \mathbb{M}^\perp)$ such that $\tau_n^2 \Psi^2(\bar{\mathbb{M}}) \leq \frac{\kappa}{128}$, we have*

$$||\hat{\theta} - \theta^*||^2 \leq \varepsilon_n^2(\bar{\mathbb{M}}, \mathbb{M}^\perp) \tag{19}$$

**Remark.** Theorem 3.3 shows that the convergence rates of the error $||\hat{\theta} - \theta^*||^2$ (8) in (Wainwright, 2019) still hold for any stationary points. The main results (Theorem 1) of (Loh & Wainwright, 2015) build on the class of nonconvex regularizers and nonconvex loss functions. Specially, they focus on the nonconvex SCAD and MCP regularizers, while our Theorem 3.3 is based on the decomposable regularizers. Moreover, Theorem 1 of (Loh & Wainwright, 2015) uses $\ell_1$ and $\ell_2$ norms to measure the error, while our Theorem 3.3 applies for any norms.

*Proof.* Let $\hat{\Delta} = \hat{\theta} - \theta^*$. Combining the RSC condition (10) and first-order necessary condition (9) yields

$$\kappa/2||\hat{\Delta}||^2 - \tau_n^2 \Phi^2(\hat{\Delta})$$
$$\leq \langle \nabla \mathcal{L}_n(\hat{\theta}) - \nabla \mathcal{L}_n(\theta^*), \hat{\Delta} \rangle$$
(10)
$$= \langle \nabla \mathcal{L}_n(\hat{\theta}), \hat{\Delta} \rangle - \langle \nabla \mathcal{L}_n(\theta^*), \hat{\Delta} \rangle$$
$$\leq \langle \lambda_n \nabla \Phi(\hat{\theta}), \theta^* - \hat{\theta} \rangle - \langle \nabla \mathcal{L}_n(\theta^*), \hat{\Delta} \rangle$$
(14)
$$\leq \lambda_n(\Phi(\theta^*) - \Phi(\hat{\theta})) - \langle \nabla \mathcal{L}_n(\theta^*), \hat{\Delta} \rangle$$
(the convexity of $\Phi$)
$$\leq \lambda_n(\Phi(\hat{\Delta}_{\bar{\mathbb{M}}}) + 2\Phi(\theta^*_{\mathbb{M}^\perp}) - \Phi(\hat{\Delta}_{\bar{\mathbb{M}}^\perp})) - \langle \nabla \mathcal{L}_n(\theta^*), \hat{\Delta} \rangle$$
(Lemma 3.2)
$$\leq \frac{3\lambda_n}{2}\Phi(\hat{\Delta}_{\bar{\mathbb{M}}}) + 2\lambda_n \Phi(\theta^*_{\mathbb{M}^\perp}) \tag{20}$$

where the final step of (20) uses the Hölder's inequality, assumed bound $\Phi^*(\nabla \mathcal{L}_n(\theta^*)) \leq \lambda_n/2$ and the triangle inequality. (7) implies that

$$\Phi^2(\hat{\Delta}) \leq 32\Psi^2(\bar{\mathbb{M}})||\hat{\Delta}||^2 + 32\Phi^2(\theta^*_{\mathbb{M}^\perp}) \tag{21}$$

The definition 2.4 of subspace Lipschitz constant implies that

$$\Phi(\hat{\Delta}_{\bar{\mathbb{M}}}) \leq \Psi(\bar{\mathbb{M}})||\hat{\Delta}|| \tag{22}$$

Combining (20), (21) and (22), we have

$$(\kappa/2 - 32\tau_n^2 \Psi^2(\bar{\mathbb{M}}))||\hat{\Delta}||^2 - \frac{3\lambda_n}{2}\Psi(\bar{\mathbb{M}})||\hat{\Delta}||$$
$$- (32\tau_n^2 \Phi^2(\theta^*_{\mathbb{M}^\perp}) + 2\lambda_n \Phi(\theta^*_{\mathbb{M}^\perp})) \leq 0 \tag{23}$$

using the assumed bound $32\tau_n^2\Psi^2(\bar{\mathbb{M}}) \leq \kappa/4$, we have

$$
\begin{aligned}
&\frac{\kappa}{4}||\hat{\Delta}||^2 - \frac{3\lambda_n}{2}\Psi(\bar{\mathbb{M}})||\hat{\Delta}|| \\
&- (32\tau_n^2\Phi^2(\theta_{\mathbb{M}^\perp}^*) + 2\lambda_n\Phi(\theta_{\mathbb{M}^\perp}^*)) \leq 0
\end{aligned}
\tag{24}
$$

Note that the left-hand side of (24) is a 2-degree polynomial in $||\hat{\Delta}||$. To be non-positive, it requires (19) to hold, which concludes the proof. $\square$

The remainder of this paper is devoted to illustrations of the consequences of Theorem 3.3 for various nonconvex loss functions.

# 4. Examples

This section illustrates the application of Theorem 3.3 to two nonconvex models: corrected linear regression and $\ell_1$-penalized Lasso estimator.

## 4.1. Corrected Linear Regression

We consider the case of high-dimensional linear regression with systematically corrupted observations. Suppose that we observe a collection of $n$ samples $(x_i, y_i) \in \mathbb{R}^d \times \mathbb{R}$. We consider the following linear model

$$
y_i = \langle x_i, \theta^* \rangle + \epsilon_i, \quad \text{for} \quad i = 1, 2, \ldots, n. \tag{25}
$$

where the regression vector $\theta^* \in \mathbb{R}^d$ is unknown and $\epsilon_i \in \mathbb{R}$ is observation noise, independent of $x_i$. A line of past work (Rosenbaum & Tsybakov, 2010; Loh & Wainwright, 2012) assume that we instead observe pairs $\{(z_i, y_i)\}_{i=1}^n$, where the $z_i$'s are systematically corrupted versions of the corresponding $x_i$'s. This setup applies to various corruption mechanisms, including the additive noise: We observe $z_i = x_i + w_i$, where $w_i \in \mathbb{R}^d$ is a random vector independent of $x_i$, say zero-mean with known covariance matrix $\Sigma_w$. We consider a high-dimensional framework that allows the feature dimensions $d$ to grow and possibly exceed the sample size $n$.

We denote the transpose of the vector/matrix by the superscript $'$. $||\cdot||_r$ represents the $\ell_r$ norm ($r \geq 1$). Let $\Sigma_x$ be the covariance matrix of the covariates, and consider the $\ell_1$-regularized quadratic program

$$
\hat{\theta} \in \arg\min_{\theta \in \mathbb{R}^d}\{\frac{1}{2}\theta'\Sigma_x\theta - \langle\Sigma_x\theta^*, \theta\rangle + \lambda_n||\theta||_1\} \tag{26}
$$

In practice, we may not know the covariance matrix $\Sigma_x$ and $\Sigma_x\theta^*$. Given a set of samples $\{(z_i, y_i)\}_{i=1}^n$, we propose to use $\hat{\Gamma} \in \mathbb{R}^{d\times d}$ and $\hat{\gamma} \in \mathbb{R}^d$ to estimate $\Sigma_x$ and $\Sigma_x\theta^*$, respectively. The corrected linear regression estimator is given by

$$
\hat{\theta} \in \arg\min_{\theta \in \mathbb{R}^d}\{\frac{1}{2}\theta'\hat{\Gamma}\theta - \langle\hat{\gamma}, \theta\rangle + \lambda_n||\theta||_1\} \tag{27}
$$

(27) involves different choices of the pair $(\hat{\Gamma}, \hat{\gamma})$ that are adapted to the cases of noisy and/or missing data.

For the case of noisy or missing data, the most natural choice of the matrix $\hat{\Gamma}$ is not positive semidefinite, hence the quadratic loss appearing in the problem (27) is nonconvex. Furthermore, when $\hat{\Gamma}$ has negative eigenvalues, the objective in (27) is unbounded from below. Hence, we make use of the following regularized estimator:

$$
\hat{\theta} \in \arg\min_{||\theta||_1 \leq B}\{\frac{1}{2}\theta'\hat{\Gamma}\theta - \langle\hat{\gamma}, \theta\rangle + \lambda_n||\theta||_1\} \tag{28}
$$

Let $y = (y_1, \ldots, y_n)' \in \mathbb{R}^n$ and $X \in \mathbb{R}^{n\times d}$, with $x_i'$ as its $i$-th row. Suppose we observe $Z = X + W$, where $W$ is a random matrix independent of $X$, with rows $w_i$ drawn independent and identically distributed (i.i.d.) from a zero-mean distribution with known covariance $\Sigma_w$. (Loh & Wainwright, 2012) consider the pair

$$
\hat{\Gamma} = \frac{Z'Z}{n} - \Sigma_w, \quad \text{and} \quad \hat{\gamma} = \frac{Z'y}{n}. \tag{29}
$$

Note that when $\Sigma_w = 0$ (corresponding to the noiseless case), the estimators (29) reduce to the standard Lasso. When $\Sigma_w \neq 0$, the matrix $\hat{\Gamma}$ is not positive semidefinite in the high-dimensional regime. Since the matrix $\frac{Z'Z}{n}$ has rank at most $n$, the subtracted matrix $\Sigma_w$ may cause $\hat{\Gamma}$ to have a large number of negative eigenvalues. Let $\lambda_{min}(\Sigma_x)$ be the minimum eigenvalue of matrix $X$.

**Definition 4.1.** We say that a random matrix $X \in \mathbb{R}^{n\times d}$ is sub-Gaussian with parameters $(\Sigma, \sigma^2)$ if:

1. each row $x_i' \in \mathbb{R}^d$ is sampled independently from a zero-mean distribution with covariance $\Sigma$, and

2. for any unit vector $u \in \mathbb{R}^d$, the random variable $u'x_i$ is sub-Gaussian with parameter at most $\sigma$.

(Wainwright, 2019) have shown the decomposability of the $\ell_1$ norm regularizer. The following theorem shows the estimation error bound for any stationary point $\hat{\theta}$ of the nonconvex program (28).

**Theorem 4.2.** *Suppose we have i.i.d. observations $\{(z_i, y_i)\}_{i=1}^n$ from a corrupted linear model with additive noise, where the random matrices $X, W \in \mathbb{R}^{n\times d}$ are sub-Gaussian with parameters $(\Sigma_x, \sigma_x^2)$ and $(\Sigma_w, \sigma_w^2)$, respectively. And let $\epsilon$ be an i.i.d. sub-Gaussian vector with parameter $\sigma_\epsilon$. Let $\sigma_z^2 = \sigma_w^2 + \sigma_x^2$. Suppose the true regression vector $\theta^*$ is supported on a subset $S$ of cardinality $s$. Assume $n \geq c_0's\max(\frac{(\sigma_x^2+\sigma_w^2)^2}{\lambda_{min}^2(\Sigma_x)}, 1)\log d$, $\lambda_n = c_0B(\sigma_x^2 + \sigma_z\sigma_\epsilon + \sigma_w\sigma_x + \sigma_z^2 + ||\Sigma_x||_2)\sqrt{\frac{\log d}{n}}$. Any stationary point $\hat{\theta}$ of the nonconvex program (28) satisfies*

*the estimation error bounds*

$$||\hat{\theta} - \theta^*||$$
$$\leq \frac{c_0 B\sqrt{s}(\sigma_x^2 + \sigma_z\sigma_\epsilon + \sigma_w\sigma_x + \sigma_z^2 + ||\Sigma_x||_2)\sqrt{\log d}}{\lambda_{min}(\Sigma_x)\sqrt{n}} \tag{30}$$

*with probability at least* $1 - c_1 \exp(-c_2 n \min(\frac{\lambda_{min}^2(\Sigma_x)}{(\sigma_x^2 + \sigma_w^2)^2}, 1)) - c_1 \exp(-c_2 \log d)$.

**Remark.** The bounds of Theorem 4.2 agree with bounds previously established in Theorem 1 of (Loh & Wainwright, 2012) and Corollary 1 of (Loh & Wainwright, 2015).

**Lemma 4.3** (RSC condition). *Under the conditions of Theorem 4.2, there are universal positive constants $c_i$ such that the loss function satisfies the RSC condition (10) with parameters $\kappa = \lambda_{min}(\Sigma_x)$ and $\tau_n^2 = c_0'\lambda_{min}(\Sigma_x)\max(\frac{(\sigma_x^2 + \sigma_w^2)^2}{\lambda_{min}^2(\Sigma_x)}, 1)\frac{\log d}{n}$*

$$\Delta'\hat{\Gamma}\Delta \geq \kappa/2||\Delta||_2^2 - \tau_n^2||\Delta||_1^2$$

*with probability at least* $1 - c_1 \exp(-c_2 n \min(\frac{\lambda_{min}^2(\Sigma_x)}{(\sigma_x^2 + \sigma_w^2)^2}, 1))$.

*Proof.* The loss function of (28) is $\mathcal{L}_n(\theta) := \frac{1}{2}\theta'\hat{\Gamma}\theta - \langle\hat{\gamma}, \theta\rangle$. So we have $\nabla\mathcal{L}_n(\theta) = \hat{\Gamma}\theta - \hat{\gamma}$, and $\langle\nabla\mathcal{L}_n(\theta^* + \Delta) - \nabla\mathcal{L}_n(\theta^*), \Delta\rangle = \Delta'\hat{\Gamma}\Delta$. The conclusion follows easily from Lemma 1 in Appendix A of (Loh & Wainwright, 2012). $\square$

The following Lemma in (Loh & Wainwright, 2012) plays the key role in the remainder of our proof.

**Lemma 4.4.** *If $X \in \mathbb{R}^{n \times d_1}$ is a zero-mean sub-Gaussian matrix with parameters $(\Sigma_x, \sigma_x^2)$, then for any fixed (unit) vector $v \in \mathbb{R}_1^d$, we have*

$$\mathbb{P}(|||Xv||_2^2 - E||Xv||_2^2| \geq nt)$$
$$\leq 2\exp(-cn\min(\frac{t^2}{\sigma_x^4}, \frac{t}{\sigma_x^2})) \tag{31}$$

*Moreover, if $Y \in \mathbb{R}^{n \times d_2}$ is a zero-mean sub-Gaussian matrix with parameters $(\Sigma_y, \sigma_y^2)$, then*

$$\mathbb{P}(||\frac{Y'X}{n} - cov(y_i, x_i)||_\infty \geq t)$$
$$\leq 6d_1 d_2 \exp(-cn\min(\frac{t^2}{\sigma_x^2\sigma_y^2}, \frac{t}{\sigma_x\sigma_y})) \tag{32}$$

*where $y_i$ and $x_i$ are the $i$-th rows of $X$ and $Y$, respectively. In particular, if $n \geq c' \log d$, then*

$$\mathbb{P}(||\frac{Y'X}{n} - cov(y_i, x_i)||_\infty \geq c_0\sigma_x\sigma_y\sqrt{\frac{\log d}{n}}) \tag{33}$$
$$\leq c_1 \exp(-c_2 \log d))$$

**Lemma 4.5** (Dual norm bound). *Under the conditions of Theorem 4.2, there are universal positive constants $c_i$ such that*

$$||\hat{\gamma} - \hat{\Gamma}\hat{\theta}||_\infty \leq \lambda_n/2$$

*with probability at least* $1 - c_1 \exp(-c_2 \log d)$, *where* $\lambda_n = c_0 B(\sigma_x^2 + \sigma_z\sigma_\epsilon + \sigma_w\sigma_x + \sigma_z^2 + ||\Sigma_x||_2)\sqrt{\frac{\log d}{n}}$.

*Proof.* Combining $\hat{\Gamma} = \frac{Z'Z}{n} - \Sigma_w$, $\hat{\gamma} = \frac{Z'y}{n}$ and $y = X\theta^* + \epsilon$, we have

$$\Phi^*(\nabla\mathcal{L}_n(\hat{\theta}))$$
$$= ||\hat{\gamma} - \hat{\Gamma}\hat{\theta}||_\infty$$
$$= ||\frac{Z'(X\theta^* + \epsilon)}{n} - (\frac{Z'Z}{n} - \Sigma_w)\hat{\theta}||_\infty$$
$$= ||\frac{Z'X\theta^*}{n} + \frac{Z'\epsilon}{n} - (\frac{Z'Z}{n} - \Sigma_z + \Sigma_z - \Sigma_w)\hat{\theta}||_\infty$$
$$= ||\frac{X'X\theta^*}{n} + \frac{W'X\theta^*}{n} + \frac{Z'\epsilon}{n} - (\frac{Z'Z}{n} - \Sigma_z)\hat{\theta}$$
$$- (\Sigma_z - \Sigma_w)\hat{\theta}||_\infty$$
$$= ||\frac{X'X\theta^*}{n} - \Sigma_x\theta^* + \frac{W'X\theta^*}{n} + \frac{Z'\epsilon}{n}$$
$$- (\frac{Z'Z}{n} - \Sigma_z)\hat{\theta} + \Sigma_x(\theta^* - \hat{\theta})||_\infty$$
$$\leq ||\frac{X'X\theta^*}{n} - \Sigma_x\theta^*||_\infty + ||\frac{W'X\theta^*}{n}||_\infty$$
$$+ ||\frac{Z'\epsilon}{n}||_\infty + ||(\frac{Z'Z}{n} - \Sigma_z)\hat{\theta}||_\infty + ||\Sigma_x(\theta^* - \hat{\theta})||_\infty \tag{34}$$

For the first term in the right hand of (34), we note that $cov(x_i, x_i\theta^*) = \Sigma_x\theta^*$. Assume $n \geq c' \log d$, applying (33) with $(X, Y) = (X, X\theta^*)$, we have

$$\mathbb{P}(||\frac{X'X\theta^*}{n} - \Sigma_x\theta^*||_\infty \geq c_0\sigma_x^2||\theta^*||_2\sqrt{\frac{\log d}{n}}) \tag{35}$$
$$\leq c_1 \exp(-c_2 \log d)$$

Similarly, we have

$$\mathbb{P}(||\frac{W'X\theta^*}{n}||_\infty \geq c_0\sigma_x\sigma_w||\theta^*||_2\sqrt{\frac{\log d}{n}})$$
$$\leq c_1 \exp(-c_2 \log d)$$
$$\mathbb{P}(||\frac{Z'\epsilon}{n}||_\infty \geq c_0\sigma_z\sigma_\epsilon\sqrt{\frac{\log d}{n}})$$
$$\leq c_1 \exp(-c_2 \log d) \tag{36}$$
$$\mathbb{P}(||(\frac{Z'Z}{n} - \Sigma_z)\hat{\theta}||_\infty \geq c_0\sigma_z^2||\hat{\theta}||_2\sqrt{\frac{\log d}{n}})$$
$$\leq c_1 \exp(-c_2 \log d)$$

Using the Hölder's inequality, we have

$$||\Sigma_x(\theta^* - \hat{\theta})||_\infty \le ||\Sigma_x(\theta^* - \hat{\theta})||_2 \le ||\Sigma_x||_2||\theta^* - \hat{\theta}||_2 \tag{37}$$

Setting $B = ||\theta^*||_1$, then we have $||\theta^*||_2 \le ||\theta^*||_1 = B$ and $||\hat{\theta}||_2 \le ||\hat{\theta}||_1 \le B$. (34), (35), (36) and (37) imply the results. $\square$

*Proof.* (of Theorem 4.2). We prove the bound (30) via an application of Theorem 3.3. (Wainwright, 2019) have shown the decomposability of the $\ell_1$ norm regularizer with subspaces

$$\mathbb{M} = \bar{\mathbb{M}} := \{\theta \in \mathbb{R}^d | \theta_j = 0 \quad \text{for all} \quad j \in S^c\}$$

$$\mathbb{M}^\perp := \{\theta \in \mathbb{R}^d | \theta_j = 0 \quad \text{for all} \quad j \in S\}$$

With this choice, note that we have $\Psi^2(\bar{\mathbb{M}}) = s$ and the target parameter $\theta^*$ is contained within a subspace $\mathbb{M}$. Lemma 4.3 shows that the RSC condition (10) holds with high probability. Lemma 4.5 shows that the event $\tilde{\mathbb{G}}(\lambda_n)$ holds with high probability. The assumption $n \ge c_0' s \max(\frac{(\sigma_x^2 + \sigma_w^2)^2}{\lambda_{min}^2(\Sigma_x)}, 1) \log d$ implies $\tau_n^2 \Psi^2(\bar{\mathbb{M}}) \le \frac{\kappa}{128}$. Thus, we complete the proof of Theorem 4.2. $\square$

### 4.2. $\ell_1$-penalized Lasso Estimator

Suppose that we observe a collection of $n$ samples $(x_i, y_i) \in \mathbb{R}^d \times \mathbb{R}$. We consider the following Gaussian linear model

$$y = X\theta^* + \epsilon \tag{38}$$

where the regression vector $\theta^* \in \mathbb{R}^d$ is unknown, $\epsilon = (\epsilon_1, \dots, \epsilon_n)' \in \mathbb{R}^n$ is the noise and $\epsilon_i$ are drawn independent and identically distributed (i.i.d.) from Gaussian distribution $\mathcal{N}(0, \sigma^{*2})$ with zero mean and standard deviation $\sigma^*$. The $\ell_1$-penalized Lasso estimator is defined as

$$\hat{\theta} \in \arg\min_{\theta \in \mathbb{R}^d}\{||y - X\theta||_2^2/2n + \lambda_n||\theta||_1\} \tag{39}$$

We can see that the standard Lasso estimator (39) does not provide an estimate of the nuisance parameter $\sigma^{*2}$.

Having a good estimator of $\sigma^{*2}$ plays a vital part in mixture models and the role of the scaling with the variance parameter is much more important than in homogeneous regression models. Hence, it is important to take $\sigma^{*2}$ into the definition and optimization of the penalized maximum likelihood estimator. We consider the following estimator:

$$(\hat{\theta}, \hat{\sigma}) \in \arg\min_{||\theta||_1 \le B, \sigma}\{\log(\sigma) + ||y - X\theta||_2^2/2n + \lambda_n||\theta||_1\} \tag{40}$$

Note that the loss function in (40) is non-convex.

We know that the $\ell_1$ norm regularizer is decomposable. The following theorem shows the estimation error bound for any stationary point $(\hat{\theta}, \hat{\sigma})$ of the nonconvex program (40).

**Theorem 4.6.** *Suppose we have i.i.d. observations $\{(x_i, y_i)\}_{i=1}^n$ from a Gaussian linear model, where the random matrix $X \in \mathbb{R}^{n \times d}$ is sub-Gaussian with parameters $(\Sigma_x, \sigma_x^2)$. Suppose the true regression vector $\theta^*$ is supported on a subset $S$ of cardinality $s$. Assume $n \ge c_0' s \max(\frac{\sigma_x^2}{\lambda_{min}(\Sigma_x)}, 1) \log d$, $\lambda_n = c_0 B(\sigma_x^2 + \sigma_x\sigma^* + ||\Sigma_x||_2)\sqrt{\frac{\log d}{n}}$. Any stationary point $(\hat{\theta}, \hat{\sigma})$ of the nonconvex program (40) satisfies the estimation error bounds*

$$||(\hat{\theta} - \theta^*, \hat{\sigma} - \sigma^*)||$$
$$\le \frac{c_0 B\sqrt{s}(\sigma_x^2 + \sigma_x\sigma^* + ||\Sigma_x||_2)\sqrt{\log d}}{\lambda_{min}(\Sigma_x)\sqrt{n}} \tag{41}$$

*with probability at least $1 - c_1 \exp(-c_2 n \min(\frac{\lambda_{min}(\Sigma_x)}{\sigma_x^2}, 1)) - c_1 \exp(-c_2 \log d)$.*

**Lemma 4.7** (RSC condition). *Under the conditions of Theorem 4.6, there are universal positive constants $c_i$ such that the loss function satisfies the RSC condition (10) with parameters $\kappa = \lambda_{min}(\Sigma_x)$ and $\tau_n^2 = c_0' \lambda_{min}(\Sigma_x) \max(\frac{\sigma_x^2}{\lambda_{min}(\Sigma_x)}, 1)\frac{\log d}{n}$*

$$\frac{1}{n}\Delta_1' X'X\Delta_1 - \frac{\Delta_2^2}{(\sigma^* + \Delta_2)\sigma^*} \ge \kappa/2||\Delta||_2^2 - \tau_n^2||\Delta||_1^2$$

*with probability at least $1 - c_1 \exp(-c_2 n \min(\frac{\lambda_{min}(\Sigma_x)}{\sigma_x^2}, 1))$.*

*Proof.* The loss function of (40) is $\mathcal{L}_n(\theta, \sigma) := \log(\sigma) + ||y - X\theta||_2^2/2n$. So we have $\nabla\mathcal{L}_n(\theta, \sigma) = (-X'(y - X\theta)/n, 1/\sigma)$. Let $\Delta = (\Delta_1, \Delta_2)$ where $\Delta_1 \in \mathbb{R}^d$ and $\Delta_2 \in \mathbb{R}^+$. We have

$$\langle \nabla\mathcal{L}_n(\theta^* + \Delta_1, \sigma^* + \Delta_2) - \nabla\mathcal{L}_n(\theta^*, \sigma^*), \Delta \rangle$$
$$= \frac{1}{n}\Delta_1' X'X\Delta_1 - \frac{\Delta_2^2}{(\sigma^* + \Delta_2)\sigma^*} \tag{42}$$

From Lemma 1 in Appendix A of (Loh & Wainwright, 2012), we know that

$$\frac{1}{n}\Delta_1' X'X\Delta_1$$
$$\ge \frac{\lambda_{min}(\Sigma_x)}{2}||\Delta_1||_2^2 \tag{43}$$
$$- c_0\lambda_{min}(\Sigma_x) \max(\frac{\sigma_x^2}{\lambda_{min}(\Sigma_x)}, 1)\frac{\log d}{n}||\Delta_1||_1^2$$

holds with probability at least $1 - c_1 \exp(-c_2 n \min(\frac{\lambda_{min}(\Sigma_x)}{\sigma_x^2}, 1))$. Let $\kappa = \lambda_{min}(\Sigma_x)$

and $\Upsilon = c_0 \lambda_{min}(\Sigma_x) \max(\frac{\sigma_x^2}{\lambda_{min}(\Sigma_x)}, 1) \frac{\log d}{n}$. The following inequality

$$\frac{1}{n}\Delta_1' X' X \Delta_1 - \frac{\Delta_2^2}{(\sigma^* + \Delta_2)\sigma^*}$$

$$\geq \frac{\kappa}{2}||\Delta_1||_2^2 - \Upsilon ||\Delta_1||_1^2 - \frac{\Delta_2^2}{(\sigma^* + \Delta_2)\sigma^*}$$

$$\geq \frac{\kappa}{2}||\Delta_1||_2^2 + \frac{\kappa}{2}\Delta_2^2 - \frac{\kappa}{2}\Delta_2^2 - \frac{\Delta_2^2}{(\sigma^* + \Delta_2)\sigma^*} - \Upsilon ||\Delta||_1^2$$

$$= \frac{\kappa}{2}||\Delta||_2^2 - (\frac{\kappa}{2} + \frac{1}{(\sigma^* + \Delta_2)\sigma^*})\Delta_2^2 - \Upsilon ||\Delta||_1^2$$

$$\geq \frac{\kappa}{2}||\Delta||_2^2 - (\frac{\kappa}{2} + \frac{1}{(\sigma^* + \Delta_2)\sigma^*})||\Delta||_1^2 - \Upsilon ||\Delta||_1^2$$

$$\geq \frac{\kappa}{2}||\Delta||_2^2 - (\frac{\kappa}{2} + \frac{1}{\sigma^{*2}} + \Upsilon)||\Delta||_1^2$$

(44)

holds with probability at least $1 - c_1 \exp(-c_2 n \min(\frac{\lambda_{min}(\Sigma_x)}{\sigma_x^2}, 1))$. $\square$

**Lemma 4.8** (Dual norm bound). *Under the conditions of Theorem 4.6, there are universal positive constants $c_i$ such that*

$$||(-X'(y - X\hat{\theta})/n, 1/\hat{\sigma})||_\infty \leq \lambda_n/2$$

*with probability at least $1 - c_1 \exp(-c_2 \log d)$, where $\lambda_n = c_0 B(\sigma_x^2 + \sigma_x \sigma^* + ||\Sigma_x||_2)\sqrt{\frac{\log d}{n}}$.*

*Proof.* Using $y = X\theta^* + \epsilon$, we have

$$\Phi^*(\nabla \mathcal{L}_n(\hat{\theta}, \hat{\sigma}))$$
$$= ||(-X'(y - X\hat{\theta})/n, 1/\hat{\sigma})||_\infty$$
$$= ||(-X'(X\theta^* + \epsilon)/n + X'X\hat{\theta}/n, 1/\hat{\sigma})||_\infty$$
$$= ||(X'X(\hat{\theta} - \theta^*)/n - X'\epsilon/n, 1/\hat{\sigma})||_\infty$$
$$= ||((X'X/n - \Sigma_x)(\hat{\theta} - \theta^*) + \Sigma_x(\hat{\theta} - \theta^*)$$
$$- X'\epsilon/n, 1/\hat{\sigma})||_\infty$$
$$\leq \max\{||(X'X/n - \Sigma_x)(\hat{\theta} - \theta^*)||_\infty + ||X'\epsilon/n||_\infty$$
$$+ ||\Sigma_x(\hat{\theta} - \theta^*)||_\infty, 1/\hat{\sigma}\}$$

(45)

For the first term in the right hand of (45), we note that $cov(x_i, x_i(\hat{\theta} - \theta^*)) = \Sigma_x(\hat{\theta} - \theta^*)$. Assume $n \geq c' \log d$, applying (33) with $(X, Y) = (X, X(\hat{\theta} - \theta^*))$, we have

$$\mathbb{P}\Big(||\frac{X'X(\hat{\theta} - \theta^*)}{n} - \Sigma_x(\hat{\theta} - \theta^*)||_\infty$$
$$\geq c_0 \sigma_x^2 ||\hat{\theta} - \theta^*||_2 \sqrt{\frac{\log d}{n}}\Big) \leq c_1 \exp(-c_2 \log d)$$

(46)

Similarly, we have

$$\mathbb{P}(||\frac{X'\epsilon}{n}||_\infty \geq c_0 \sigma_x \sigma^* \sqrt{\frac{\log d}{n}}) \leq c_1 \exp(-c_2 \log d)$$

(47)

Using the Hölder's inequality, we have

$$||\Sigma_x(\theta^* - \hat{\theta})||_\infty \leq ||\Sigma_x(\theta^* - \hat{\theta})||_2 \leq ||\Sigma_x||_2||\theta^* - \hat{\theta}||_2$$

(48)

Setting $B = ||\theta^*||_1$, then we have $||\theta^*||_2 \leq ||\theta^*||_1 = B$ and $||\hat{\theta}||_2 \leq ||\hat{\theta}||_1 \leq B$. Combining (45), (46), (47) and (48) derives the results. $\square$

*Proof.* (of Theorem 4.6). We prove the bound (41) via an application of Theorem 3.3. The proof of Theorem 4.2 has shown that $\Psi^2(\bar{\mathbb{M}}) = s$ and the target parameter $\theta^*$ is contained within a subspace $\mathbb{M}$. Lemma 4.7 shows that the RSC condition (10) holds with high probability. Lemma 4.8 shows that the event $\tilde{\mathbb{G}}(\lambda_n)$ holds with high probability. The assumption $n \geq c_0' s \max(\frac{\sigma_x^2}{\lambda_{min}(\Sigma_x)}, 1) \log d$ implies $\tau_n^2 \Psi^2(\bar{\mathbb{M}}) \leq \frac{\kappa}{128}$. Thus, we complete the proof of Theorem 4.6. $\square$

## 5. Conclusion

This paper extends the theory of $M$-estimators with decomposable regularizers from convex to nonconvex. Theorem 3.1 recovers the results of the Proposition 2.2 in (Wainwright, 2019) for any stationary points. Theorem 3.3 recovers the convergence rates of the error $||\hat{\theta} - \theta^*||^2$ (8) in (Wainwright, 2019) for any stationary points. Moreover, we use two simple nonconvex examples to illustrate our main results.

## Acknowledgements

This work is supported by the Key R&D Program of Hubei Province under Grant 2024BAB038, the National Key R&D Program of China under Grant 2023YFC3604702, the Fundamental Research Funds for the Central Universities under Grant 2042025kf0045.

## Impact Statement

This paper presents work whose goal is to advance the field of Machine Learning. There are many potential societal consequences of our work, none which we feel must be specifically highlighted here.

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
