# OpenReview forum: "Nonconvex Theory of $M$-estimators with Decomposable Regularizers"
_ICML.cc/2025/Conference — ICML 2025 poster_

### Official Review · Reviewer_p2mm · 2025-03-10

**Overall Recommendation:** 4

**Summary:**

This paper challenges the results of Section 9 of Martin's textbook "High-dimensional statistics". Surprisingly, this paper is able to recover the results of Proposition 9.13 and Theorem 9.19 of "High-dimensional statistics" for nonconvex loss functions. Moreover, Theorem 3.3 in this paper extends the results of Theorem 1 in Po-Ling's nonconvex m-estimators work to the general norms. They also consider the corrected linear regression and l1-penalized Lasso estimator as two nonconvex samples

**Claims And Evidence:**

Yes

**Essential References Not Discussed:**

NA

**Experimental Designs Or Analyses:**

NA

**Methods And Evaluation Criteria:**

Yes

**Other Comments Or Suggestions:**

NA

**Other Strengths And Weaknesses:**

1. Based on the definitions of stationary points (9), RSC condition(10) and $\tilde{\mathbb{G}}(\lambda_n)$, Theorem 3.1 in this manuscript recovers the results of Proposition 9.13 of Martin's textbook for any stationary points. Theorem 3.3 in this manuscript is an important result. Because they recover the convergence rate of Theorem 9.19 of Martin's textbook for any stationary points, and improve the Po-Ling's nonconvex results to any norms. Theorem 4.2 also recovers the previous results. Theorems 3.1 and 3.3 are quite novel and of significant to the community due to the importance of nonconvex research.

2. Theorem 4.2 only deals with the sub-Gaussian parameters which may be a weakness of this paper. The DNN is the popular nonconvex structure, but the examples section did not present the DNN case.

**Questions For Authors:**

-Can we consider the other distributions for corrected linear regression?
-Can we use the DNN as the example?

**Relation To Broader Scientific Literature:**

NA

**Theoretical Claims:**

I have read the proof of this paper, and believe there are correct.

---

> ### Author Rebuttal · Authors · 2025-03-30
>
> -Can we consider the other distributions for corrected linear regression?
>
> Thanks for the question. (Rosenbaum & Tsybakov, 2010; Loh & Wainwright, 2012) have already studied the corrected linear regression, and they consider the sub-Gaussian parameters. This paper just follow them to use the exiting non-convex eample to illustrate our theory. It is interesting to consider the parameters that follow other distributions. But it is out of the scope of this paper.
>
>
> -Can we use the DNN as the example?
>
> Thanks for the question. We have tried DNN as an example. Unfortunately, DNN do not satifies the dual norm bound. Therefore, we can not apply our theory to DNN. One interesting future direction is to study the non-convex theory that can be applied to DNN.

---

### Official Review · Reviewer_UkML · 2025-03-10

**Overall Recommendation:** 3

**Summary:**

This paper studies the theoretical properties of regularized M-estimators with decomposable regularizers under nonconvex loss functions. The authors extend existing results on convex regularized M-estimators to the nonconvex case, demonstrating that estimation errors remain within a restricted set and that convergence rates can still be recovered. They establish key theoretical guarantees, leveraging restricted strong convexity and decomposability conditions. The theoretical results are further supported by two concrete applications: corrected linear regression and the Lasso estimator under nonconvex loss functions. The findings significantly contribute to the understanding of high-dimensional statistical estimation in nonconvex settings.

**Claims And Evidence:**

yes

**Essential References Not Discussed:**

no

**Experimental Designs Or Analyses:**

No experiment

**Methods And Evaluation Criteria:**

yes

**Other Comments Or Suggestions:**

NO

**Other Strengths And Weaknesses:**

Strengths
-Theoretical novelty: Extends convex M-estimator results to the nonconvex setting.
-Strong mathematical foundations: Proofs are detailed and rigorous.
-Relevance: The results have broad applications in high-dimensional statistics.
-Clarity: The paper is well-structured and clearly presents key results.

Weaknesses
-Theoretical focus: Lacks empirical validation, though this is not necessarily a major drawback given the paper's aims. -Assumptions: Some assumptions (e.g., decomposability, restricted strong convexity) may not always hold in practical settings, limiting applicability to certain problems.

**Questions For Authors:**

-Can you comment on whether your results extend to more general forms of nonconvexity beyond those studied in the examples?

**Relation To Broader Scientific Literature:**

This work extends foundational results in high-dimensional statistics, particularly those of Negahban et al. (2009), Wainwright (2019), and Loh & Wainwright (2015). While these prior works focused on convex regularized M-estimators, the present paper generalizes the theory to nonconvex loss functions.

**Theoretical Claims:**

I have checked the theories and found no errors.

---

> ### Author Rebuttal · Authors · 2025-03-30
>
> Can you comment on whether your results extend to more general forms of nonconvexity beyond those studied in the examples?
>
> Thanks for the question. Our framework applies to a broad class of nonconvex loss functions; however, we require that the nonconvex loss satifies the dual norm bound.

---

> > ### Comment · Reviewer_UkML · 2025-04-02
> >
> > My questions have been addressed. Thanks for the reply.

---

### Official Review · Reviewer_RMP7 · 2025-03-12

**Overall Recommendation:** 3

**Summary:**

This paper develops a theoretical framework for analyzing regularized M-estimators with decomposable regularizers. Extending prior work in convex settings, the authors establish that estimation errors remain in a restricted set and that convergence rates can be recovered despite the loss function's nonconvexity.

**Claims And Evidence:**

yes

**Essential References Not Discussed:**

No

**Experimental Designs Or Analyses:**

NA

**Methods And Evaluation Criteria:**

yes

**Other Comments Or Suggestions:**

The paper can benefit from a brief discussion on potential algorithmic implementations based on the theoretical results.

**Other Strengths And Weaknesses:**

Provides a significant theoretical extension from convex to nonconvex settings. Uses rigorous mathematical analysis to establish key results. Well-structured and clearly written, making the contributions accessible. Offers practical examples to illustrate the main theoretical findings. I think there is no significant weakness.

**Questions For Authors:**

Are there any specific classes of nonconvex loss functions where your framework does not apply?
How sensitive are the key theoretical results to violations of the decomposability assumption?

**Relation To Broader Scientific Literature:**

This paper is close to Wainwright (2019) and Loh & Wainwright (2015).

**Theoretical Claims:**

yes

---

> ### Author Rebuttal · Authors · 2025-03-30
>
> 1, The paper can benefit from a brief discussion on potential algorithmic implementations based on the theoretical results.
>
> Thanks for the question. Our theoretical results show that the decomposable regularizers play the key role in facilitating convergence and improving generalization. So, it is important to develop the regularizers for the potential algorithmic implementations.
>
> 2, Are there any specific classes of nonconvex loss functions where your framework does not apply? How sensitive are the key theoretical results to violations of the decomposability assumption?
>
> Thanks for the question. Our framework applies to a broad class of nonconvex loss functions; however, we require that the nonconvex loss satifies the dual norm bound. If the nonconvex loss does not satisfy the dual norm bound, our results do not hold. The decomposability assumption plays a crucial role in our results. If this assumption is violated, the key bounds and statistical guarantees may no longer hold.

---

### Official Review · Reviewer_fa9A · 2025-03-13

**Overall Recommendation:** 5

**Summary:**

The paper studies the high dimensional M-estimators for non-convex loss functions. The previous classical results only consider the convex cases. It is natural to consider the non-convex loss function in high dimensions. The motivation is strong. The central theoretical questions studied in this paper are whether we can extend the classical results from convex to non-convex cases. The paper shows the positive results which are quite interesting and the proof is easy to follow.

**Claims And Evidence:**

The claims made in this paper are well-supported by rigorous mathematical derivations and proofs.

**Essential References Not Discussed:**

No

**Experimental Designs Or Analyses:**

No

**Methods And Evaluation Criteria:**

The methods employed in the paper are mathematically rigorous and appropriate for the problem setting.

**Other Comments Or Suggestions:**

No

**Other Strengths And Weaknesses:**

Strengths

Originality

The paper studies the important theoretical questions which are never covered by previous works. This paper is the original research.

Quality

The decomposable regularizers and restricted strong convexity are two key concepts that play the important roles in the high dimension statistics. The decomposable regularizers enforce that the estimated errors fall into a restricted set. Motivated by this, the restricted strong convexity is defined to show the property of strong convexity only along some directions. The classical results show that one can get the desired convergence rates of the estimated error. The basic assumptions are the convexity of loss functions. The paper aims to break the convex assumption and consider the non-convex settings. To address the questions, the paper is mainly based on the weaker RSC condition (Loh & Wainwright, 2015) than (4). The main results in this paper show that both the Proposition 2.2 and convergence rate of the estimated error in (Wainwright, 2019) still hold for any stationary points. The proof is simple and rigorous. The paper also use two non-convex examples to illustrate the theories. The overall quality is quite impressive.

Clarity

The motivation, the background and the main proofs are well organized and easy to follow.

Significance

The main results of this paper are important. They extend our knowledge on high dimensional M-estimators from classical convex to nonconvex cases. The significance is that they may motivate the other researchers to study more challenging non-convex loss functions.

Weaknesses

Although the results obtained this paper are important, I still have some questions for the authors:
1, What is the main difference between the proof of Theorem 2.7 in (Wainwright, 2019) and Theorem 3.3 in this paper? I think it is better to clarify this question in the paper, then the reader can quickly understand theorems and proofs.
2, What is the difference between the $\tilde{\mathbb{G}}(\lambda_n)$ defined in this paper and $\mathbb{G}(\lambda_n)$ defined in  (Wainwright, 2019)? Why not use the original definition of $\mathbb{G}(\lambda_n)$ in  (Wainwright, 2019)?
3, The first example is corrected linear regression. Why corrected linear regression model is non-convex and why do we need to enforce the constraint for parameter $\theta$?

**Questions For Authors:**

See the Weaknesses.

**Relation To Broader Scientific Literature:**

(Loh & Wainwright, 2015) present the nonconvex results of regularized M-estimator with non-convex regularizers. This paper considers the nonconvex results of regularized M-estimator with decomposable regularizers.

**Theoretical Claims:**

I carefully reviewed key results, including Theorem 3.1 and Theorem 3.3. The derivations are logically sound. The proofs appear correct, assuming the stated assumptions hold.

---

> ### Author Rebuttal · Authors · 2025-03-30
>
> 1, What is the main difference between the proof of Theorem 2.7 in (Wainwright, 2019) and Theorem 3.3 in this paper? I think it is better to clarify this question in the paper, then the reader can quickly understand theorems and proofs.
>
> Thanks for the question. The main difference between the proof of Theorem 2.7 in (Wainwright, 2019) and Theorem 3.3 in this paper is that the proof of Theorem 2.7 in (Wainwright, 2019) uses the RSC condition (4), while our proof of Theorem 3.3 relies on the  RSC condition (10). The difference between  (4) and (10) contributes the key difference between Theorem 2.7 in (Wainwright, 2019) and Theorem 3.3 in this paper.
>
>
> 2, What is the difference between the $\tilde{\mathbb{G}}(\lambda_n)$ defined in this paper and $\mathbb{G}(\lambda_n)$ defined in (Wainwright, 2019)? Why not use the original definition of $\mathbb{G}(\lambda_n)$ in  (Wainwright, 2019)?
>
> Thanks for the question.  The $\tilde{\mathbb{G}}(\lambda_n)$ defined in this paper is for stationary points, while the $\mathbb{G}(\lambda_n)$ defined in  (Wainwright, 2019) is for the unknown parameter. The price for recovering the Proposition 2.2 in  (Wainwright, 2019) to nonconvex settings is that we have to redefine the $\tilde{\mathbb{G}}(\lambda_n)$ for stationary points.
>
> 3, The first example is corrected linear regression. Why corrected linear regression model is non-convex and why do we need to enforce the constraint for parameter $\theta$?
>
> Thanks for the question.  For the case of noisy or missing data, the most natural choice of the matrix $\hat{\Gamma}$ is not positive semidefinite, hence the quadratic loss appearing in the problem is nonconvex.  (Rosenbaum & Tsybakov, 2010; Loh & Wainwright,
> 2012) have already shown the nonconvexity of this problem. Furthermore, when $\hat{\Gamma}$ has negative eigenvalues, the objective in (27) is unbounded from below. Hence, we  enforce the constraint for parameter $\theta$.  (Loh & Wainwright,
> 2012) have already done like this.

---

> > ### Comment · Reviewer_fa9A · 2025-04-02
> >
> > Thanks author for the response. They have answered my questions well. I keep my positive score.

---

### Decision · Program_Chairs · 2025-05-01

**Decision:**

Accept (poster)

**Comment:**

This submission studies the nonconvex theory of M-estimator through decomposable regularizers. The key point that the authors take a step forward in this area is extending the assumption of the convexity of the loss function to the high-dimensional non-convex. The authors provide several principled insights behind this relaxation.

This submission received the comments of four reviewers, who respectively recommended 5, 3, 3, 4. After the rebuttal, two reviewers confirmed the authors' feedback while the other two reviewers only acknowledged without further comments. AC has checked the review and the comments and feel some comments are too short and fall out of consideration. But based on the detailed review of the first reviewer and the last reviewer, AC feel at least there are no negative problems about this submission.

In this case, based on the valid reviewers' suggestion, AC tend to recommend "Acceptance" towards this submission.